

# Fleas of black rats (*Rattus rattus*) as reservoir host of *Bartonella* spp. in Chile

Lucila Moreno Salas[1], Mario Espinoza-Carniglia[1],
Nicol Lizama Schmeisser[1], L. Gonzalo Torres[1,2],
María Carolina Silva-de la Fuente[3,4], Marcela Lareschi[5] and
Daniel González-Acuña[3]

[1] Departamento de Zoología, Facultad de Ciencias Naturales y Oceanográficas, Universidad de Concepción, Concepción, Chile
[2] Facultad de Ciencias, Programa de Magíster en Ciencias mención Ecología Aplicada, Universidad Austral de Chile, Valdivia, Chile
[3] Departamento de Ciencia Animal, Facultad de Ciencias Veterinarias, Laboratorio de Parásitos y Enfermedades de Fauna Silvestre, Universidad de Concepción, Chillán, Chile
[4] Facultad de Medicina Veterinaria, Universidad San Sebastián, Concepción, Chile
[5] Centro de Estudios Parasitológicos y de Vectores CEPAVE (CONICET CCT-La Plata-UNLP), La Plata, Argentina

Corresponding author
Lucila Moreno Salas,
lumoreno@udec.cl

## ABSTRACT

**Background:** *Rattus rattus* is a widely distributed, invasive species that presents an important role in disease transmission, either directly or through vector arthropods such as fleas. These black rats can transmit a wide variety of pathogens, including bacteria of the genus *Bartonella*, which can cause diseases in humans and animals. In Chile, no data are available identifying fleas from synanthropic rodents as *Bartonella* vectors. The aim of this study was to investigate the prevalence of *Bartonella* spp. in the fleas of *R. rattus* in areas with different climate conditions and featuring different human population densities.

**Methods:** In all, 174 fleas collected from 261 *R. rattus* captured from 30 localities with different human densities (cities, villages, and wild areas) across five hydrographic zones of Chile (hyper-arid, arid, semi-arid, sub-humid, and hyper-humid) were examined. *Bartonella* spp. presence was determined through polymerase chain reaction, using *gltA* and *rpoB* genes, which were concatenated to perform a similarity analysis with BLAST and phylogenetic analysis.

**Results:** Overall, 15 fleas species were identified; *Bartonella gltA* and *rpoB* fragments were detected in 21.2% (37/174) and 19.5% (34/174) of fleas, respectively. A total of 10 of the 15 fleas species found were positive for *Bartonella* DNA. *Leptopsylla segnis* was the most commonly collected flea species ($n = 55$), and it also presented a high prevalence of *Bartonella* DNA ($P\% = 34.5\%$). The highest numbers of fleas of this species were collected in villages of the arid zone. There were no seasonal differences in the prevalence of *Bartonella* DNA. The presence of *Bartonella* DNA in fleas was recorded in all hydrographic areas, and the arid zone presented the highest prevalence of this species. Regarding areas with different human densities, the highest prevalence was noted in the villages (34.8% *gltA* and 31.8% *rpoB*), followed by cities (14.8% *gltA* and 11.1% *rpoB*) and wild areas (7.4% *gltA* and 14.8% *rpoB*). The BLAST analysis showed a high similitude (>96%) with four uncharacterized *Bartonella* genotypes and with two species with zoonotic potential: *B. mastomydis* and *B. tribocorum*. The phylogenetic analysis showed a close relationship with

*B. elizabethae* and *B. tribocorum*. This is the first study to provide evidence of the presence of *Bartonella* in fleas of *R. rattus* in Chile, indicating that the villages and arid zone correspond to areas with higher infection risk.

## INTRODUCTION

*Bartonella* spp. are vector-borne bacteria that have been identified in a wide range of mammals (*Breitschwerdt, 2017*). Among these, rodents are described as important reservoirs of *Bartonella* (*Ying et al., 2002*; *Favacho et al., 2015*; *Gonçalves et al., 2016*). Of the 45 species named to date, 35 have been registered in rodents and/or fleas, of which 13 have been identified as potentially pathogenic to humans (*Chomel et al., 2009*; *Jiyipong et al., 2014*; *Alsarraf et al., 2017*), and five have been implicated in different infections in humans (*Daly et al., 1993*; *Kosoy et al., 2003*; *Serratrice et al., 2003*; *Fenollar, Sire & Raoult, 2005*; *Buffet, Kosoy & Vayssier-Taussat, 2013*).

Although reports of human transmission are not frequent, some recent studies support the possibility that rodent-associated *Bartonella* species may be responsible for human infections, especially in areas where humans and rats are in close contact; these infections are most prevalent in homeless people and are more likely to be contracted while engaging in outdoor activities (*Kosoy et al., 2008*, *2010*; *Ying et al., 2012*). In several of these infection cases, fleas were recognized as the vectors or potential vectors of these bacteria (*Chomel et al., 2009*); as such, fleas are believed to play a key role in maintaining the *Bartonella* species in rodents (*Buffet, Kosoy & Vayssier-Taussat, 2013*; *Billeter et al., 2014*), although the role they could play in human infections is unknown.

*Rattus rattus* (black rat) has been identified as a *Bartonella* reservoir in different areas of the world (*Ellis et al., 1999*; *Hsieh et al., 2010*; *Pangjai et al., 2014*; *Bai et al., 2009*; *Gonçalves et al., 2016*; *Peterson et al., 2017*). *Rattus rattus* is widely distributed in most areas of the world due to human movement (*Krystufek et al., 2016*) and it has been cataloged as the most harmful invasive species in the world, as it has caused the extinction and displacement of several species of birds and mammals; it is also considered one of the main disease vectors for humans and wild animals (*Banks & Hughes, 2012*; *Harris, 2009*; *Towns, Atkinson & Daugherty, 2006*). The latter fact is due to the rat's close contact with humans, as they live in cities and rural areas, and have been able to colonize wild environments, interacting with native species (*Lobos, Ferres & Palma, 2005*). Thus, the *Bartonella* species in *R. rattus* may be the result of a host switching between native species (*Ellis et al., 1999*).

In Chile, only five *Bartonella* species have been described in domestic animals and humans: *B. rochalimae* in the human flea *Pulex irritans* (*Pérez-Martínez et al., 2009*), *B. koehlerae*, *B. clarridgeiae*, and *B. henselae* in cat blood (*Ferrés et al., 2005*; *Zaror et al., 2002*; *Müller et al., 2017*); these last two species have also been detected in cat fleas

(*Pérez-Martínez et al., 2009*), and no species have been detected in rodents and/or the fleas associated with them. In humans, there are reports of a high prevalence of *B. henselae* in children, veterinarians, and zookeepers (*Ferrés et al., 2005*; *Troncoso et al., 2016*), and reports of infection-related diseases with *B. henselae* and *B. quintana* in people (*Uribe et al., 2012*; *Sandoval et al., 2014*; *Zepeda et al., 2016*; *Arce, González & Madrid, 2017*).

Chile presents a contrasting diversity of climates due to its long extension (between −35.675148 and −71.5429688), with regions ranging from deserts to rainforests (*CONAMA, 2008*). *Rattus rattus* has been able to colonize many of these environments (*Lobos, Ferres & Palma, 2005*; *Iriarte, 2007*), which present seasonal changes that can affect the presence and density of hosts and vectors, and can impact the prevalence of *Bartonella* (*Telfer et al., 2007*; *Friggens et al., 2010*). Due to the close contact that black rats maintain with humans and wild species; the high number of flea species described for rats in Chile (12 species; *Beaucournu, Moreno & González-Acuña, 2014*), and how the fleas that parasitize them can act as potential *Bartonella* vectors; as well as the scarce knowledge that exists about *Bartonella* in Chile associated with human synanthropic rodents, we investigated the prevalence of *Bartonella* spp. in *R. rattus* fleas in areas characterized by different human population densities throughout the different hydrographic zones and seasons in Chile.

## MATERIALS AND METHODS

### Sample localities and rodent-trapping procedure

This study surveyed rodents and fleas in 30 localities (cities, *n* = 10; village, *n* = 10; wild areas, *n* = 10) of five hydrographic zones (hyper-arid, arid, semi-arid, sub-humid, and hyper-humid, between −20.2167 and −53.1667 lat.; Fig. 1) of Chile from December 2015 to January 2018. This study took place during austral summer (December to February) and austral winter (July and September), except in the hyper-arid and hyper-humid hydrographic zones, which were visited only in the winter and summer, respectively. The sample localities were chosen based on the following demographic characteristics: (1) City: an urban entity with more than 5,000 inhabitants; (2) Village: an urban entity with a population ranging from 2,001 to 5,000 inhabitants or between 1,001 and 2,000 people and met the economic activity requirement (*INE, 2005*); and (3) wild area: a protected area without human settlement; for the latter, permission was requested from the Corporación Nacional Forestal (CONAF N°018-2015).

The rodents were captured using live cage traps baited with oats. Each locality was sampled for two consecutive nights. In each sampling locality, the traps were placed in four parallel lines (approximately 100 m from each other) and each line was equipped with 50 traps (with a distance of 10 m between each other), with a total sampling effort of 12,000 traps per night. The rodents were removed from the traps according to standard techniques (*Mills et al., 1995*). Each animal was identified using the description by *Iriarte (2007)*, anesthetized with ketamine:xylazine (1:1), and euthanized by cervical dislocation (*American Veterinary Medical Association, 2013*). The carcasses were placed in individual bags with 95% alcohol and transported to the laboratory. Animal use was

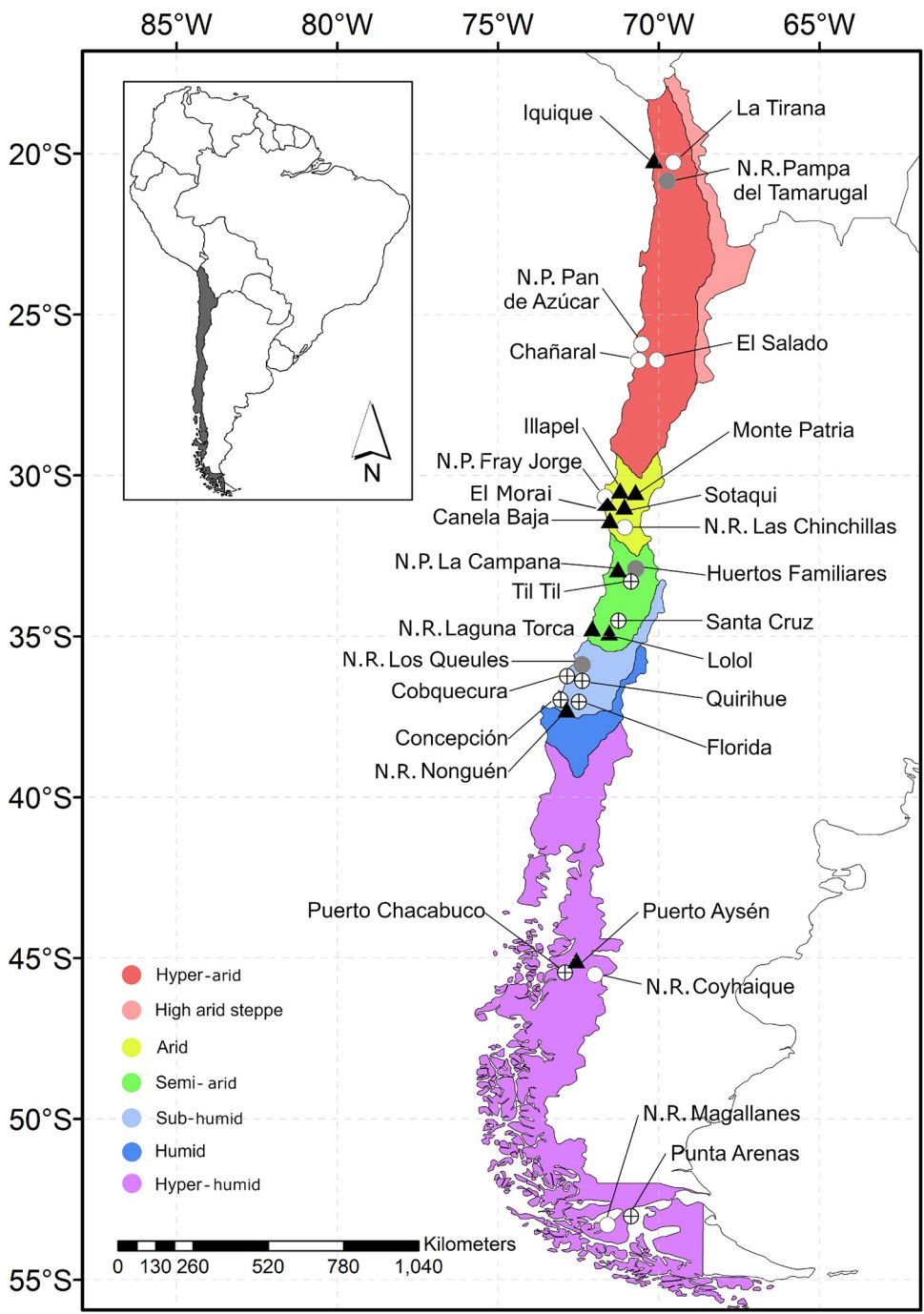

**Figure 1 Map of Chile indicating the location of the study sites.** Each data point indicates sample locality. Gray circle: locality featuring rats without fleas; circle with a cross: locality featuring rats with fleas; black triangle: locality featuring fleas that tested positive for *Bartonella* DNA; white circle: locality without rats.                           

conducted in accordance with the protocols for field and laboratory rodent studies (*Herbreteau et al., 2011*), and the protocols were approved by the Comité de Ética de la Vicerrectoría de Investigación y Desarrollo de la Universidad de Concepción.

**Table 1 Primer sequences used for PCR amplifications.**

| Name | Primer | Product length (bp) |
|---|---|---|
| BaGlta_F | TCTACGGTACGTCTTGCTGGATCA | 201 |
| BaGlta_R | GCCCATAAGGCGGAAAGGATCATT | 201 |
| BaRpoB_F | CGCGCGATCATGTTGATTTGATGG | 159 |
| BaRpoB_R | ATGGTGCTTCAGCACGTACAAGAG | 159 |

**Note:**
F, forward; R, reverse.

## Fleas: sample collection

Each rodent was placed on a white plastic basin and the fleas were collected immediately in the field. The rodents' fur was thoroughly brushed with a toothbrush and the fleas were collected by hand or with forceps, and stored individually in sterile cryovials with 95% ethanol. Later, in the laboratory, the carcasses were checked to verify that all fleas had been collected. For each rodent, the total number of fleas extracted was recorded (abundance) and with this data, the mean intensity of infection (the number of fleas collected from all species/number of infested hosts), the mean abundance (MA) of infection (the number of collected fleas from all species/total number of hosts), and prevalence (the number of infected hosts) were calculated.

## DNA extraction

For DNA extraction, every flea was washed and cut between the third and fourth abdominal tergites with a sterile scalpel. The material used to handle the fleas was sterilized between each sample. DNA was extracted using a commercial kit (Qiagen®, Hilden, Germany) according to the manufacturer's instructions. The incubation time was 5 h, after which point a final elution step was performed using 200 μL of AE buffer and stored at −20 °C.

Following the DNA extraction, the fleas' exoskeletons were recovered and stored in 96% ethanol; there were subsequently mounted for fleas' species identification. DNA contamination was monitored by an extraction control using distilled water, every 10 samples.

## PCR amplification of *gltA* and *rpoB* genes

The presence of *Bartonella* was screened using the citrate synthase (*gltA*) and RNA polymerase beta-subunit (*rpoB*) genes. The primers used for DNA amplification and sequencing in this study were designed from a partial *gltA* and *rpoB* gene sequence of *B. tribocorum* (GenBank code: AM260525.1; Table 1).

For the amplification of the *gltA* gene fragment, the polymerase chain reaction (PCR) program was modified with an initial denaturation for 5 min at 95 °C, followed by 40 cycles (95 °C for 30 s, 56.2 °C for 30 s, and 72 °C for 30 s), and a final extension step at 72 °C for 5 min. For the *rpoB* gene, the PCR was started by denaturation for 5 min in 95 °C, followed by 40 cycles (95 °C for 30 s, 56.6 °C for 30 s, and 72 °C for 30 s), and a final extension step at 72 °C for 5 min. Reactions were performed in 26 μL of mixture

containing GoTaq® Green Master Mix (Promega Corporation, Madison, WI, USA) 2× 12.5 μL + 5.5 μL of free ultrapure water nuclease + two μL of forward primer (10 μM) + two μL of reverse primer (10 μM) + four μL of DNA sample. Negative controls for the PCR consisted of a blank DNA extraction and distilled water was added to the PCR mix instead of DNA. Positive control was the genomic DNA of *Bartonella henselae* (Vircell Microbiologist, Granada, Spain). PCR products were subjected to electrophoresis on 1% agarose gel at a voltage of 100 V. Then, the PCR products from positive samples were sequenced by Macrogen Company (Seoul, South Korea).

## Sequencing, BLAST, and phylogenetic analysis

The DNA sequences used in this study and the known *Bartonella* species retrieved from GenBank were aligned using Codon Code Aligner (Codon Code Corporation; Files S1 and S2). The sequencing data of *gltA* and *rpoB* were concatenated and compared with the sequences of *Bartonella* available in GenBank using the nucleotide–nucleotide BLAST (blastn) program (see http://www.ncbi.nlm.nih.gov/BLAST/). A substitution saturation test with DAMBE (*Xia, 2017*) was performed, showing that the sequences were not saturated (*Xia et al., 2003*; *Xia & Lemey, 2009*). We used MEGA7 (*Kumar, Stecher & Tamura, 2016*) to calculate the genetic distances between sequences. A tree with Bayesian probabilities was computed using MrBayes 3.2 (*Ronquist et al., 2012*) based on concatenated *gltA* (142 bp) and *rpoB* (95 bp) gene fragments, using *Brucella abortus* as an outgroup (Accession number, *gltA*: LIUE01000001.1; *rpoB*: CP023241.1). The GTR substitution model was used to reconstruct the tree and perform 10,000,000 bootstrap trials. Haplotype diversity (Hd), segregating sites (S), and nucleotide diversity (π) were calculated using DNAsp 6. The accession numbers of the GenBank sequences used to reconstruct the tree are detailed in Fig. 2.

## Mounting fleas

Fleas were mounted on glass slides using conventional procedures (*Hastriter & Whiting, 2003*). Fleas were identified at the species level using the taxonomic keys and the descriptions of *Hopkins & Rothschild (1956*, *1962*, *1966)*, *Smit (1987)*, *Schramm (1987)*, *Beaucournu, Torres Mura & Gallabdo (1988)*, *Beaucournu & Gallardo (1988)*, *Beaucournu & Kelt (1990)*, *Beaucournu, Moreno & González-Acuña (2011)*, *Sánchez et al. (2012)*, and *Sánchez & Lareschi (2014)*. Voucher specimens of each flea species were deposited in the specimen repository of the Museo de Zoología, Universidad de Concepcion, Concepción, Chile (MZU-CCCC-46329–46336).

## Data analysis

The prevalence (*P*%), MA, and mean infestation intensity (MI) of fleas was calculated according to *Rózsa, Reiczigel & Majoros (2000)* and were compared between seasons, hydrographic zones, and areas with different population densities using chi-squared and Fisher's exact tests to compare prevalence, while a Bootstrap *t*-test with 2,000 replications was used to compare MA and MI. The Clopper–Pearson test was used to calculate the confidence interval (CI) for prevalence, and a Bootstrap test was used to calculate the CI for

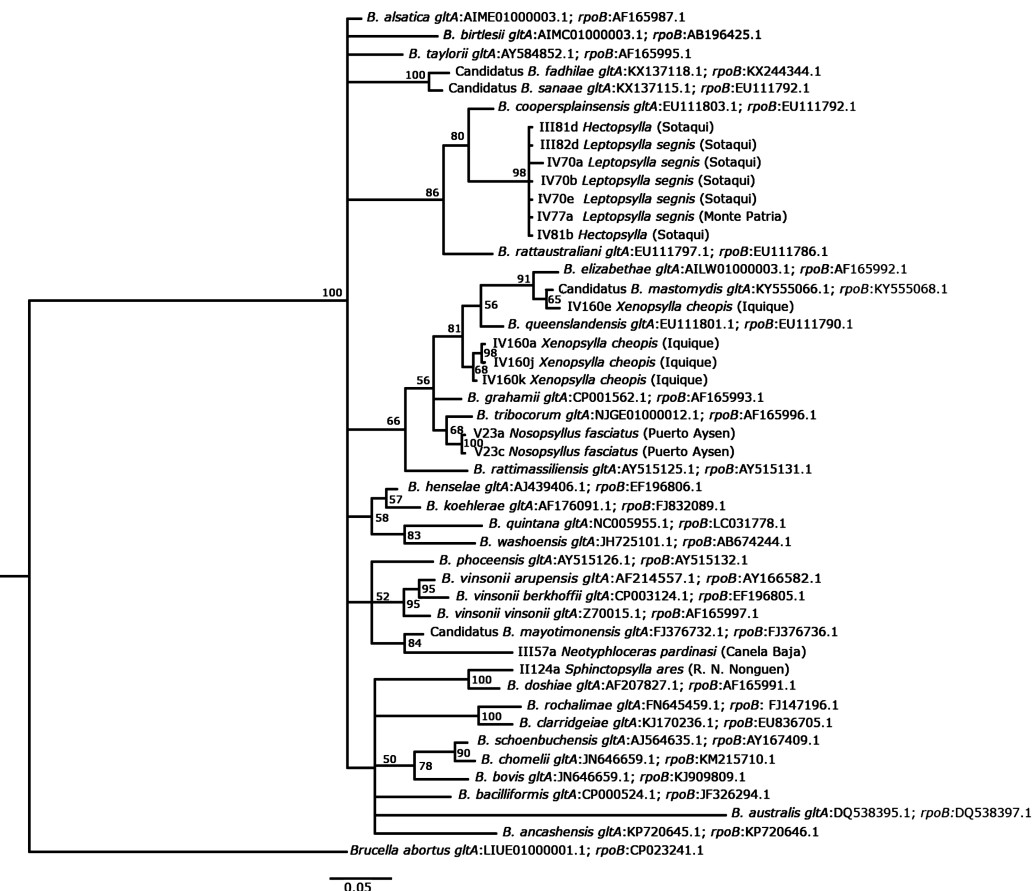

**Figure 2 Phylogenetic tree of *Bartonella*, as based on concatenated *glt*A and *rpo*B genes using a GTR substitution model.** The phylogenetic tree was constructed using a Bayesian method. *Brucella abortus* was included as an outgroup. Bootstrap values were calculated with 10,000,000 replicates. The corresponding accession number for each genotype is indicated below each species of *Bartonella*. The flea species from which *Bartonella* DNA was detected is indicated, and the locality and the location from where it was collected is noted in parentheses.

MI and MA. *Bartonella* prevalence (percentage of infected fleas) was calculated based on the PCR results. The associations between *Bartonella* infection and hydrographic zone, human density, and season were evaluated using the chi-squared test, and for small sample sizes, F-Fisher was used. A *P*-value less than 0.05 was considered to be statistically significant. Statistical analyses were performed using the Quantitative Parasitology software (QP 3.0; *Rózsa, Reiczigel & Majoros, 2000*).

## Nucleotide sequence accession numbers

The sequences of *Bartonella* gene fragments generated in this study were deposited in the NCBI GenBank database under the following accession numbers: *gltA*: MK720786–MK720800, and *rpoB*: MK720801–MK720815.

## RESULTS

A total of 261 *R. rattus* (summer: *n* = 139; winter: *n* = 122) were collected in 21 of the 30 localities (city: *n* = 149; village: *n* = 53; wild area: *n* = 59) and 31% (*n* = 81) of the black

**Table 2 Detection of *Bartonella* DNA from fleas collected on *Rattus rattus* from different seasons and locality types.**

| Family | Specie of flea | Total of fleas analyzed by seasons | | Total of fleas analyzed by type of locality | | | Number of fleas positive for gene fragment | |
|---|---|---|---|---|---|---|---|---|
| | | Summer | Winter | City | Village | Reserve | *glt*A | *rpo*B |
| Pulicidae | *Xenopsylla cheopis* | 0 | 11 | 11 | 0 | 0 | 7 (63.6) | 5 (45.5) |
| Leptopsyllidae | *Leptopsylla segnis* | 22 | 33 | 19 | 36 | 0 | 19 (34.5) | 15 (27.3) |
| Ceratophyllidae | *Nosopsyllus fasciatus* | 19 | 26 | 33 | 12 | 0 | 4 (8.9) | 4 (8.9) |
| Hectopsyllidae | *Hectopsylla* sp. | 3 | 0 | 0 | 3 | 0 | 3 (100) | 2 (66.7) |
| Hystrichopsyllidae | *Ctenoparia inopinata* | 1 | 0 | 1 | 0 | 0 | 0 | 0 |
| | *Ctenoparia jordani* | 0 | 1 | 0 | 0 | 1 | 0 | 0 |
| | *Neotyphloceras* sp. | 4 | 0 | 1 | 3 | 0 | 0 | 1 (25.0) |
| | *Neotyphloceras chilensis* | 2 | 0 | 0 | 2 | 0 | 1 (50.0) | 1 (50.0) |
| | *Neotyphloceras pardinasi* | 14 | 9 | 9 | 6 | 8 | 1 (4.3) | 3 (13.0) |
| Rhopalopsyllidae | *Delostichus coxalis* | 6 | 0 | 0 | 0 | 6 | 0 | 1 (16.7) |
| | *Delostichus smiti* | 0 | 1 | 1 | 0 | 0 | 0 | 0 |
| | *Tetrapsyllus rhombus* | 1 | 2 | 0 | 1 | 2 | 1 (33.3) | 0 |
| Stephanocircidae | *Plocopsylla* sp. | 0 | 2 | 1 | 0 | 1 | 0 | 0 |
| | *Plocopsylla wolffsohni* | 0 | 1 | 0 | 1 | 0 | 0 | 0 |
| | *Sphinctopsylla ares* | 3 | 13 | 5 | 2 | 9 | 1 (6.2) | 2 (12.5) |
| *Total* | | 75 | 99 | 81 | 66 | 27 | 37 (21.2) | 34 (19.5) |

rats were positive for fleas. A total of 174 fleas were collected (winter: $n = 99$; summer: $n = 75$), representing 15 species from 10 different genera and seven families (Table 2). *Ctenoparia jordani* Smit, 1955; *Neotyphloceras chilensis* (Jordan, 1936); *Neothphloceras pardinasi* Sánchez & Lareschi, 2014; *Delostichus smiti* Jameson & Fulk, 1977; *Tetrapsyllus rhombus* Smit, 1955; *Plocopsylla wolffsohni* (Rothschild, 1909); and *Sphinctopsylla ares* (Rothschild, 1911) are new records for *R. rattus* in Chile. The MI was 2.18 (range: 1–15) fleas per black rat. In nine cities, between one and five species of fleas were collected, in eight villages between one and four, and in three wild areas between three and four species of fleas. The hydrographic zones (i.e., hyper-arid and hyper-humid zones) demonstrated the lowest richness of flea species, with one and three species, respectively, while the semi-arid and sub-humid zones presented eight species each. The flea species *L. segnis* (Schönherr, 1811) ($n = 55$) and *Nosopsyllus fasciatus* (Bosc d'Antic, 1800) ($n = 45$) were the most abundant species in villages and cities, respectively, and were not found in wild area. Two species of fleas were found only in wild areas: *D. coxalis* (Rothschild, 1909) and *C. jordani*. *Hectopsylla* sp. and *Neotyphloceras chilensis* were collected only in villages. *Ctenoparia inopinata* Rothschild, 1909, *D. smiti*, and *Xenopsylla cheopis* Glinkiewicz, 1907 were exclusive to cities. *Neotyphloceras pardinasi* and *S. ares* were found in cities, villages, and wild areas (Table 2).

Although the number of captured rodents ($n = 149$) and collected fleas ($n = 111$) was higher in cities than in villages (rodents: $n = 53$; fleas: $n = 96$) and wild areas (rodents: $n = 59$; fleas: $n = 38$), the prevalence of fleas was significantly higher in the villages (45.3%, chi-squared = 6.679, d$f$ = 2, $P$ = 0.035), while wild areas (30.5%) and cities (26.2%)

**Table 3 Prevalence, mean abundance, and mean intensity of fleas, as well as the prevalence of *Bartonella* DNA from black rats captured from five hydrographic zones and 21 localities in Chile.**

| Hydrographic zone | Locality | Number rodent | % Prevalence of fleas [95% CI] | Abundance mean of fleas [95% CI] | Intensity mean of fleas [95% CI] | Number of fleas analyzed | Number of fleas positive to *gltA* (%) | Number of fleas positive *rpoB* (%) |
|---|---|---|---|---|---|---|---|---|
| Hyper-arid | Iquique[C] | 2 | 50.0 [0.01–0.99] | 5.5 [0.00–5.50] | 11.0* | 11 | 7 (63.6) | 5 (71.4) |
| | N.P. Pampa del Tamarugal[W] | 10 | 0.0 | 0.0 | – | 0 | 0 (0.0) | 0 (0.0) |
| Arid | Illapel[C] | 17 | 76.5 [0.50–0.93] | 3.2 [1.53–4.00] | 4.2 [2.15–5.00] | 33 | 2 (6.1) | 1 (3.0) |
| | Monte Patria[C] | 14 | 42.9 [0.13–0.65] | 0.9 [0.14–1.14] | 2.0 [1.00–2.20] | 7 | 1 (14.3) | 1 (14.3) |
| | El Morai[V] | 8 | 25.0 [0.03–0.65] | 0.5 [0.00–1.50] | 2.0 [1.00–2.00] | 4 | 0 (0.0) | 0 (0.0) |
| | Canela Baja[V] | 6 | 50.0 [0.12–0.88] | 1.8 [0.17–4.67] | 3.7 [1.00–5.67] | 8 | 2 (25.0) | 7 (87.5) |
| | Sotaquí[V] | 14 | 85.7 [0.57–0.98] | 4.8 [2.57–7.21] | 5.6 [3.25–8.17] | 40 | 21 (52.5) | 13 (32.5) |
| Semi-arid | Til Til[C] | 3 | 33.3 [0.01–0.91] | 0.3 [0.00–0.67] | 1.0* | 1 | 0 (0.0) | 0 (0.0) |
| | Santa Cruz[C] | 5 | 20.0 [0.01–0.72] | 0.2 [0.00–0.40] | 1.0* | 0 | 0 (0.0) | 0 (0.0) |
| | Huertos Familiares[V] | 2 | 0.0 | 0.0 | – | 0 | 0 (0.0) | 0 (0.0) |
| | Lolol[V] | 1 | 100.0 [0.25–1.00] | 4.0* | 4.0* | 4 | 0 (0.0) | 1 (25.0) |
| | N.P. La Campana[W] | 22 | 22.7 [0.08–0.45] | 0.6 [0.18–1.14] | 2.6* | 13 | 1 (7.7) | 1 (7.7) |
| | N.R. Laguna Torca[W] | 2 | 50.0 [0.01–0.99] | 2.5 [0.00–2.50] | 5.0* | 5 | 0 (0.0) | 2 (40.0) |
| Sub-humid | Quirihue[C] | 77 | 13.0 [0.06–0.22] | 0.2 [0.09–0.40] | 1.6 [1.10–2.40] | 15 | 0 (0.0) | 0 (0.0) |
| | Concepción[C] | 22 | 22.7 [0.08–0.45] | 0.4 [0.09–0.77] | 1.6 [1.00–2.20] | 8 | 0 (0.0) | 0 (0.0) |
| | Cobquecura[V] | 1 | 100.0 [0.02–1.00] | 2.0* | 2.0* | 2 | 0 (0.0) | 0 (0.0) |
| | Florida[V] | 18 | 22.2 [0.64–0.48] | 0.3 [0.06–0.78] | 1.5 [1.00–2.00] | 6 | 0 (0.0) | 0 (0.0) |
| | N.R. Nonguén[W] | 25 | 48.0 [0.28–0.69] | 0.8 [0.44–1.36] | 1.7 [1.17–2.58] | 9 | 1 (11.1) | 1 (11.1) |
| Hiper-humid | Puerto Aysén[C] | 4 | 25.0 [0.01–0.80] | 1.5 [0.0–3.0] | 6.0* | 5 | 2 (40.0) | 2 (40.0) |
| | Punta Arenas[C] | 5 | 20.0 [0.01–0.72] | 0.2 [0.0–0.4] | 1.0* | 1 | 0 (0.0) | 0 (0.0) |
| | Puerto Chacabuco[V] | 3 | 33.3 [0.01–0.91] | 0.7 [0.0–1.33] | 2.0* | 2 | 0 (0.0) | 0 (0.0) |

**Notes:**
C, city; V, village; W, wild area; N.P., national park; N.R., national reserve.
* One specimen of *R. rattus* was captured or only one individual was positive to fleas, the confidentiality intervals could not be determined.

did not show significant differences (chi-squared = 0.399, d$f$ = 1, $P$ = 0.528). The MI was higher in villages (MI = 2.83), than in cities (MI = 2.26) and wild areas (MI = 1.89), but no statistically significant differences were found (Bootstrap $P$-value (two-sided) > 0.05). The MA was also higher in villages (MA = 1.28) than in the wild area (MA = 0.57) and cities (MA = 0.40; Bootstrap $t$-test $P$ < 0.05), but there were no significant differences between cities and wild areas (Bootstrap $t$-test $P$ = 0.9360; Table 3).

Seasonally, the prevalence, MA, and MI of fleas were higher in winter ($P$% = 34.4%, MA = 0.90, MI = 2.62) than in summer ($P$% = 28.1%, MA = 0.57, MI = 2.05), but no statistically significant differences were found between seasons ($P$%: chi-squared statistic = 1.231, d$f$ = 1, $P$ = 0.267; MA: Bootstrap $t$-test $P$ = 0.0870; MI: Bootstrap $t$-test $P$ = 0.1340; Table 4). In winter, only prevalence was significantly higher in the villages ($P$% = 57.89%, MA = 1.789, MI = 3.091) than in the cities ($P$% = 28.8%, MA = 0.881, MI = 3.059; $P$%: chi-squared statistic = 5.282, d$f$ = 1, $P$ = 0.022; AM: Bootstrap $t$-test $P$ = 0.0755; MI: Bootstrap $t$-test $P$ = 0.9660) and wild areas ($P$% = 31.8%, MA = 0.545, MI = 1.714; $P$%: chi-squared statistic = 3.77, d$f$ = 1, $P$ = 0.052; MA: Bootstrap $t$-test $P$ = 0.0155;

**Table 4 Prevalence of the *Bartonella* species in fleas collected from different localities and seasons in Chile.**

| Season | Type of locality | Number rodent collected | % Prevalence of fleas [95% Cl] | Intensity mean [95% Cl] | Number fleas analyzed | Number of fleas positive to *gltA* (%) | Number of fleas positive *rpoB* (%) |
|---|---|---|---|---|---|---|---|
| Winter | City | 59 | 28.81 [0.18–0.42] | 3.06 [2.12–4.13] | 49 | 9 (18.36) | 6 (12.24) |
| | Village | 19 | 57.89 [0.33–0.80] | 3.09 [2.27–3.91] | 33 | 15 (45.45) | 6 (18.18) |
| | Wild area | 44 | 31.80 [0.19–0.47] | 1.77 [1.00–2.57] | 17 | 2 (11.76) | 3 (17.65) |
| Total | | 122 | 34.43 [0.26–0.43] | 2.62 [2.10–3.31] | 99 | 26 (26.26) | 15 (15.15) |
| Summer | City | 90 | 24.44 [0.16–0.35] | 1.64 [1.27–2.23] | 32 | 3 (9.37) | 3 (9.37) |
| | Village | 34 | 38.20 [0.22–0.56] | 2.62 [1.77–3.62] | 33 | 8 (24.24) | 15 (44.45) |
| | Wild area | 15 | 26.67 [0.08–0.55] | 2.50 [1.00–3.00] | 10 | 0 | 1 (10) |
| Total | | 139 | 28.06 [0.21–0.36] | 2.05 [1.64–2.49] | 75 | 11 (14.67) | 19 (25.33) |

MI: Bootstrap *t*-test $P = 0.0360$). Regarding the hydrographic zones, in winter, the highest prevalence, mean intensity, and MA were found in the arid zone ($P\% = 64.3\%$, MA = 2.79, MI = 4.33), and these values were significantly different from those of the sub-humid zone ($P\% = 27.7\%$, MA = 2.78, MI = 1.78; $P\%$: chi-squared statistic = 11.045, d$f$ = 1, $P = 0.001$; MA: Bootstrap *t*-test $P = 0.011$; MI: Bootstrap *t*-test $P = 0.038$), while with the semi-arid zone, the prevalence and MA differed significantly ($P\%$: chi-squared statistic = 5.841, d$f$ = 1, $P = 0.016$; MA: Bootstrap *t*-test $P = 0.0235$), but this was not the case for mean intensity (Bootstrap *t*-test $P = 0.22$). There was no significant difference between the sub-humid ($P\% = 27.7\%$, MA = 0.492, MI = 1.78) and semi-arid zones ($P\% = 27.8\%$, MA = 0.788, MI = 2.8; $P\%$: Fisher's exact test $P = 1.00$; MA: Bootstrap *t*-test $P = 0.467$, MA: Bootstrap *t*-test $P = 0.254$).

In the summer, there were no significant differences in the prevalence (chi-squared statistic = 2.341, d$f$ = 2, $P = 0.310$), MA (Bootstrap *t*-test $P = 0.077$), and MI (Bootstrap *t*-test $P = 0.089$) of fleas between cities, villages, and wild areas (Table 4). The arid zone showed a higher prevalence, MA, and MI ($P\% = 54.8\%$, MA = 1.645, MI = 3.00) than the semi-arid ($P\% = 23.5\%$, MA = 0.588, MI = 2.51) and sub-humid zones ($P\% = 17.9\%$, MA = 0.256, MI = 1.43), although it only differed significantly in terms of prevalence with the semi-arid zone ($P\%$: chi-squared statistic = 4.373, d$f$ = 1, $P = 0.037$; MA: Bootstrap *t*-test $P = 0.045$, MI: Bootstrap *t*-test $P = 0.5635$), and in terms of prevalence and MA with the sub-humid zone ($P\%$: chi-squared statistic = 14.833, d$f$ = 1, $P = 0.000$; MA: Bootstrap *t*-test $P = 0.0280$, MI: Bootstrap *t*-test $P = 0.0605$). No significant differences were observed in these parameters between the semi-arid and sub-humid zones ($P\%$: Fisher's exact test $P = 0.733$, MA: Bootstrap *t*-test $P = 0.2545$, MI: Bootstrap *t*-test $P = 0.2275$).

The *Bartonella gltA* and *rpoB* fragment was detected in 21.26% (37/174) and 19.54% (34/174) of the fleas, respectively, collected from 22 different *R. rattus* individuals. Although a higher prevalence of *Bartonella* was detected using the *gltA* fragment, this finding was not statistically significant (chi-squared statistic = 0.159, d$f$ = 1, $P = 0.690$). A total of 10 of the 15 flea species found were positive for *Bartonella*. We observed the highest prevalence in *Hectopsylla* sp. (100%) and *X. cheopis* (63%), although the

**Table 5 *Bartonella* species detected with BLAST using concatenated *gltA* and *rpoB* genes, in the identified flea species collected in Chile.**

| Flea species | *Bartonella* species isolated | BLAST Sequence similarity (%) | GenBank accession number | Locality/hydrographic zone |
|---|---|---|---|---|
| *Xenopsylla cheopis* | *Bartonella* sp. B28297 | 100 | KM233489.1 | Iquique/Hyper-arid |
|  | *Bartonella mastomydis* | 100 | KY555066.1 | Iquique/Hyper-arid |
| *Leptopsylla segnis* | *Bartonella* sp. 16/40 | 97 | AY584859.1 | Sotaquí/Arid |
| *Nosopsyllus fasciatus* | *Bartonella* sp. 16/40 | 97 | AY584859.1 | Sotaquí/Arid |
|  | *Bartonella tribocorum* | 100 | HG969192.1 | Puerto Aysén/Hyper-humid |
| *Hectopsylla* sp. | *Bartonella* sp. 16/40 | 96 | AY584859.1 | Sotaquí/Arid |
| *Neotyphloceras chilensis* | *Bartonella* sp. (strain C1phy) | 99 | Z70022.1 | Canela Baja/Arid |
| *Neotyphloceras pardinasi* | *Bartonella* sp. (strain C1phy) | 99 | Z70022.1 | Canela Baja/Arid |
| *Sphinctopsylla ares* | *Uncultured Bartonella* sp. clone LBCE 10781 | 95 | KX270236.1 | Nonguén/Sub-humid |

number of flea samples collected from these species was low (Table 2). While *L. segnis* was the most commonly collected flea ($n = 55$), it also presented a high prevalence of *Bartonella* DNA (*gltA* = 34.54% and *rpoB* = 27.3%); the highest numbers of fleas of this species were collected in villages in the arid zone (Table 2).

The presence of *Bartonella* DNA in fleas was recorded in all hydrographic areas, but not in all localities. Of the hydrographic zones with more than 20 fleas analyzed, the arid zone had a greater prevalence of *Bartonella* DNA (26.1% *gltA* and 28.3% *rpoB*) than in the sub-humid zone (2.5% *gltA* and *rpoB*) for both genes (*gltA*: chi-squared = 10.103, d$f$ = 1, $P = 0.001$; *rpoB*: chi-squared = 11.371, d$f$ = 1, $P = 0.001$), whereas with the semi-arid area, the only difference was found in the prevalence with the *gltA* gene (4.3% *gltA*: chi-squared = 5.111, d$f$ = 1, $P = 0.024$; 17.4% *rpoB*: chi-squared: 1.127, d$f$ = 1, $P = 0.288$). The prevalence of *Bartonella* DNA in the semi-arid and sub-humid zones did not differ significantly (*gltA*: Fisher's exact test $P = 1.00$; *rpoB*: Fisher's exact test $P = 0.055$). The highest prevalence was noted in the villages (34.8% *gltA* and 31.8% *rpoB*), which differed significantly from that in the city (14.81%; chi-squared = 6.039, d$f$ = 1, $P = 0.014$) and the in the wild area (7.41%; chi-squared = 7.34, d$f$ = 1, $P = 0.007$) for the *gltA* gene, whereas for the *rpoB* gene, only the villages (31.82%) and cities (11.11%) were significantly different (chi-squared = 9.6, d$f$ = 1, $P = 0.002$). No significant differences were observed between the cities and wild areas (Fisher's exact test $P = 0.733$), and between the wild areas and villages (chi-squared = 2.82, d$f$ = 1, $P = 0.093$). In all, 22 rodents carried fleas that were positive for *Bartonella*. The fleas positive for *Bartonella* were extracted from 13 village rodents, five city rodents, and four wild-area rodents.

No statistically significant differences (*gltA*: chi-squared = 3.427, d$f$ = 1, $P = 0.064$; *rpoB*: chi-squared = 2.814, d$f$ = 1, $P = 0.093$) were observed in the prevalence of *Bartonella* DNA in fleas between the winter (*gltA*: 26.26%; *rpoB*: 15.15%) and summer (*gltA*: 14.67%; *rpoB*: 25.33%; Table 4).

The BLAST analyses showed similar findings for eight *Bartonella* spp. (Table 5). The genetic distances between the findings from GenBank and the sequences obtained in this study are shown in File S3. The phylogenetic analysis showed that concatenated *gltA*

and *rpoB* sequences could be related to a known *Bartonella* species (Fig. 2). It was found that *Bartonella* present in the flea species (*Hectopsylla* sp. and *L. segnis*) collected from cities and villages in arid zones are closely related to *B. cooperplainsensis*. *Bartonella* DNA detected in *X. cheopis* collected only in a city from a hyper-arid zone was closely related to *B. mastomydis* and *B. queenslandensis*. *Bartonella* DNA in *Nosopsyllus fasciatus* was closely related to *B. tribocorum*. Finally, *Bartonella* detected in *Neotyphloceas pardinasi* was related to *B. mayotimonensis*, while that detected in *S. ares* was related to *B. doshiae*. Eight haplotypes were found (Hd = 0.838, $S = 60$, $\pi = 0.095$).

## DISCUSSION

The presence of *Bartonella* DNA in *R. rattus* fleas has not been previously reported in Chile; therefore, this is the first study to report on and document the prevalence of this bacteria in fleas over a large spatial scale ($-20°$ to $-53°$ lat.) covering five hydrographic zones with differences in human density. To our knowledge, this is the first report of the detection of *Bartonella* spp. in several flea species: *Neotyphloceras chilensis*, *Neotyphloceas pardinasi*, *Neotyphloceras* sp., *D. coxalis*, *T. rhombus*, *S. ares*, and *Hectopsylla* sp., all of which parasitize native rodents of Chile. This indicates that *R. rattus* is in contact with wild-rodent populations and can act as a reservoir for and facilitator in the dispersion of these fleas—and in the *Bartonella* species detected. While the presence of *Bartonella* in *X. cheopis*, *L. segnis*, and *Nosopsyllus fasciatus* confirms the findings made by other authors in other parts of the world (*Parola et al., 2003*; *Winoto et al., 2005*; *Loftis et al., 2006*; *Reeves et al., 2007*; *Li et al., 2007*; *Tsai et al., 2010*; *Hornok et al., 2015*), the *Bartonella* reported in these three flea species is also new to Chile. Although it cannot be stated that these flea species are competent vectors of *Bartonella*, as they may have consumed *Bartonella*-infected blood from a host with bacteremia, their role as vectors of these bacteria cannot be ruled out; as such, future laboratory tests to verify their competence are necessary (*Billeter et al., 2008*).

The prevalence of *Bartonella* in the rodent fleas in our study is within the ranges documented by other authors (2.1–40.5%), values that vary with respect to the geographical area and flea species analyzed (*Loftis et al., 2006*; *Marie et al., 2006*; *Li et al., 2007*; *Bitam et al., 2012*; *Billeter et al., 2014*; *Dieme et al., 2015*; *Lipatova et al., 2015*). *Bartonella* DNA was found in several flea species with variations observed in the infection prevalence of *Bartonella* detected between flea species (4.2–100%). In five flea species collected from *R. rattus*, *Bartonella* DNA was not detected, which could be due to the low number of fleas analyzed in these species (between one and four individuals). However, *Bartonella* DNA prevalence was high in other species that were not abundant in the sample, such as *Hectopsylla* sp., *X. cheopis*, and *T. rhombus*. Each flea species was collected from a single rodent, which could be infected with *Bartonella*, which would explain the high prevalence. It is unlikely that finding *Bartonella* DNA in *Hectopsylla* sp. would pose a risk to human health, because these fleas are neosomatic and females are semipenetrating (*Linardi & De Avelar, 2014*), they stay attached to the host for long periods of time, representing little chance that it will infect humans. The high prevalence of *Bartonella* DNA reported in this study for *X. cheopis* (63.6%) would be within the ranges

reported for other parts of the world (*Billeter et al., 2011*: 95%; *Leulmi et al., 2014*: 34.7%; *Klangthong et al., 2015*: 25.8%; *Billeter et al., 2013*: 59.1%; *Dieme et al., 2015*: 6.7%). *Xenopsylla cheopis* is the most frequently occurring and abundant species to isolate from *R. rattus* (*Loftis et al., 2006*; *Christou et al., 2010*; *Guernier et al., 2014*), and is associated with the transmission of several pathogens to humans (e.g., the plague, endemic murine typhus, helminth parasites; *Farhang-Azad, Traub & Baqar, 1985*; *Bitam et al., 2006*; *Gárate et al., 2011*). Several species of *Bartonella* have been detected in *X. cheopis* (*B. elizabethae*, *B. grahamii*, *B. tribocorum*, *B. rochalimae*, *B. rattimassiliensis*, *B. queenslandensis*, and *Bartonella* sp. 1.1C; *Billeter et al., 2008*, *2011*; *Tsai et al., 2010*; *Dieme et al., 2015*), although its competence as a vector has only been determined experimentally for *B. elizabethae*, which would be eliminated through the feces (*McKee et al., 2018*). *Tetrapsyllus rhombus* was another very rare species, but which had a high prevalence of *Bartonella* DNA; there is no known history of pathogens that this flea species transmits. However, the finding in *R. rattus* provides evidence of the exchange of fleas between wild and introduced species. *Tetrapsyllus rhombus* is widely distributed in central and southern Chile, parasitizing 13 species of wild rodents of the families Cricetidae, Octodontidae, and Ctenomyidae (*Beaucournu, Moreno & González-Acuña, 2014*).

Conversely, fleas that were abundant in *R. rattus*, such as *L. segnis* and *Nosopsyllus fasciatus*, presented significant differences in the prevalence of *Bartonella* DNA. Few studies have detected *Bartonella* in *L. segnis*, with prevalence rates of 0% (0/174), 3% (1/37), and 10% (1/10; *Loftis et al., 2006*; *Li et al., 2007*, *Hornok et al., 2015*). *Leptopsylla segnis* is widely distributed, attributed to the dispersal of its hosts (rats and mice), as a result of human activity. This species is distributed in temperate zones, although in our study, it was distributed from the arid zone to the hyper-humid zone, with the greatest abundance observed in the arid zone. In our study, we did not find the species in wild areas, while there are records in Chile that indicate a high prevalence in wild species (68–82% in *Octodon degus*; *Burger et al., 2012*). The finding of *Bartonella* DNA in this flea species is important because it is an abundant species. Although this flea has rarely been reported to feed on humans (*Li & Xio, 1993*), it has the potential to transmit *Bartonella* through the skin via contamination of infected feces, as with other *Bartonellae*. *Nosopsyllus fasciatus* presents an abundance and geographical distribution similar to *L. segnis*; however, the prevalence of *Bartonella* DNA in this species was lower, with only 8% of fleas testing positive for the bacteria, although this value is within the ranges recorded by other authors (*Parola et al., 2003*: 3% (1/26); *Zurita, 2018*: from 4% to 13%). *Nosopsyllus fasciatus* and *L. segnis* exhibit a cosmopolitan distribution and live in temperate environments; in our study, these species were distributed in all zones, except in the hyper-arid zone, presenting with a greater abundance in the arid zone. *Nosopsyllus fasciatus* spends more time in the nest of its hosts than actually on them (*Bitam et al., 2010*) and it is fed fewer times per day (two to three times) compared to *L. segnis* (three to five times; *Kunitskaya et al., 1965*). This feature may decrease the likelihood with which these fleas acquire bacteria, as the feeding frequency and mobility of the fleas are important factors that influence pathogen transmission (*Laudisoit et al., 2014*). Although the prevalence of this flea was low and its transmission potential is unknown, it could be acting

as a *Bartonella* reservoir. In addition, it is considered an important flea in public health because it occasionally infects other mammals, including humans (*Pratt & Wiseman, 1962*). On the other hand, *Neotyphloceas pardinasi* was another abundant species in this study; this species was described in Argentina as parasites of Sigmodontinae rodents (*Sánchez & Lareschi, 2014*). Although the prevalence of *Bartonella* was low in this species, this is the first report of *Bartonella* DNA in *Neotyphloceas pardinasi*, and the first record of this flea species in *R. rattus* in Chile. The differences in prevalence found among the flea species analyzed may be due to the specificity of *Bartonella*, although more studies are needed to test this hypothesis.

Although the prevalence of *Bartonella* in fleas detected with the *gltA* gene was higher than in fleas with *rpoB*, this finding was not statistically significant. These genes were shown to have high discriminatory power inter-species; however, to be able to validate this species, long fragments are needed. *La Scola et al. (2003)* propose that newly encountered *Bartonella* isolates should be considered a new species if a 327-bp *gltA* fragment shares <96.0% sequence similarity with those of validated species, and if an 825-bp *rpoB* fragment shares <95.4% sequence similarity with those of validated species. Fragments shorter than those recommended by La Scola et al. were obtained (*gltA* = 142 bp and *rpoB* = 95 bp); therefore, we could not determine with certainty if *Bartonella* corresponds to a new species. Although the *glt*A and *rpoB* segments used for this analysis were short, these represent a reliable taxonomic tool for distinguishing between differences among closely related organisms (*Birtles & Raoult, 1996*).

Phylogenetic analysis based on the concatenated *gltA* and *rpoB* gene sequences identified groups close to well-known rat-associated Bartonellae: *B. coopersplainsensis*, *B. mastomydis*, *B. tribocorum*, *B. mayotimonensis*, and *B. doshiae*. BLAST analysis of concatenated sequences obtained from *X. cheopis* revealed 100% similarity with *B. mastomydis* and *Bartonella* sp. B28297. Both *Bartonella* species are found within the *B. elizabethae* complex (*Halliday et al., 2015*), but it is unknown whether they are pathogenic for humans. *Nosopsyllus fasciatu*s from the hyper-humid zone harbored a bacterium that was 100% identical to *B. tribocorum* and the phylogenetic analysis also showed a close relationship with this bacterium. Although this bacterium is associated with rodents and their ectoparasites, it has recently been described as a bacterium with pathogenic potential for humans, since it was isolated in human patients from Thailand (*Kosoy et al., 2010*) and France, causing acute febrile illnesses and nonspecific symptoms (*Vayssier-Taussat et al., 2016*). *Hectopsylla* sp., *L. segnis*, and *Nosopsyllus fasciatus* presented between 96% and 97% similarity with *Bartonella* sp. 16/40 detected in the rodent *Apodemus peninsulae* in Russia by *Mediannikov et al. (2005)*, although these authors indicated that this species could be new, as it is in an independent and well-isolated clade. In our study, the species was forming a well-differentiated clade, but it was related to *B. coopersplainsensis* (Fig. 2). *Bartonella* DNA detected in *Neotyphloceras chilensis* and *Neotyphloceas pardinasi* showed 99% similarity with *Bartonella* sp. C1phy detected in the blood of *Phyllotis* sp. in Peru. Similar results were found by *Cicuttin et al. (2019)*, who detected *Bartonella* in *Neotyphloceras crackensis* Sánchez & Lareschi 2014 in the province of Santa Cruz, Argentina, and which shared 100% similarity with *Bartonella* C1phy. In the

phylogenetic analysis, this *Bartonella* constitutes a clade with *B. mayotimonensis*. This bacterium has been recognized as a pathogen for humans and was isolated from the resected aortic valve tissue of a person with infective endocarditis in the US (*Lin et al., 2010*). *S. ares* showed a 95% similarity with uncultured *Bartonella* sp. clone LBCE 10781 detected in *Oxymycterus dasytrichus* in Brazil (*Rozental et al., 2017*), and was found to form a monophyletic clade with *B. doshiae*. This species holds pathogenic potential in humans, since it has been detected in human patients in France who had a history of being bitten by ticks (*Vayssier-Taussat et al., 2016*).

It is important to know the distribution of pathogens among the different biotic communities, since it implies that certain areas pose a higher risk of infection for humans (*Mills & Childs, 1998*). In our study, 10 of the 30 sampled locations (four cities, four villages, two wild areas), had fleas that were positive for *Bartonella* DNA; it was found that villages had a higher prevalence than cities and wild areas, which coincides with the greater MA and prevalence of fleas registered in villages, in addition to the differences in the dominant flea species in the different areas studied. For example, *L. segnis*, which had the highest prevalence of *Bartonella* DNA, was present in cities and villages but not in wild areas, and its MA and prevalence was greater in villages. According to our results, this species would constitute an important potential vector of *Bartonella*, as it is abundant in *R. rattus* and has a wide distribution in Chile, concentrating its greater abundance in cities and villages of the arid zone. In addition, parasitizing native rodent species were found (*Beaucournu, Moreno & González-Acuña, 2014*; *Burger et al., 2012*). Although the prevalence of *Bartonella* DNA was lower in wild areas, and only parasitic flea species of native Chilean rodents were collected, this result is important because it means that *R. rattus* could disperse *Bartonella* species present in fleas from wild areas to rural areas. It also highlights the presence of *Bartonella* in wild areas, which are used as recreational spaces for people, who may then become exposed to *Bartonella* infection. These findings suggest that the probability of coming into contact with fleas infected with *Bartonella* is higher in rural areas than in cities and wild areas.

Our study shows variations in the prevalence of *Bartonella* in the different hydrographic zones analyzed, and the differences could be associated to both the distribution of flea species and environmental factors. A study conducted on fleas of domestic animals from Tunisia (*Zouari et al., 2017*) found a higher prevalence of *Bartonella* in fleas from humid areas, followed by semi-arid, sub-humid, and arid regions; this bacterium was not found in the dry zone, contrary to what was found in our study. During our investigation, we found a higher prevalence of the bacterium in arid, semi-arid, and sub-humid zones. These differences could be explained by the differences in humidity and temperature in these areas, which determine the presence of certain flea species in some areas, affecting the dynamics of the vectors and their survival (*Chinga-Alayo et al., 2004*). Although in the hyper-arid zone the prevalence of fleas in rodents was low, we found only one species (*X. cheopis*), and noted that the prevalence of *Bartonella* DNA was high (6/11; 54%). On the other hand, in our study, a significantly higher prevalence of *Bartonella* DNA observed in the arid zone, as compared with the other zones,

could be linked with the higher prevalence of *L. segnis* and may also be responsible for the transmission of this pathogen.

*Bartonella* prevalence did not change between seasons. Although other studies have found seasonal differences in the prevalence of *Bartonella* in fleas, the authors attributed these differences to changes in the population dynamics of different flea species, as well as to changes in community composition, where the dominant species changes (*Telfer et al., 2007*). In our study, we did not find significant changes in the composition of flea species, abundance, or prevalence among the seasons analyzed, where the most abundant and prevalent species (*L. segnis* and *Nosopsyllus fasciatus*) remained stable. We highlight the high richness of flea species detected in our study compared to other studies (one to five species; *Loftis et al., 2006*; *Marie et al., 2006*; *Li et al., 2007*; *Reeves et al., 2007*; *Tsai et al., 2010*), which could be explained due to the wide geographical range considered in our study (20°–53° lat.) and also to the inclusion of wild areas, in contrast to other studies that only include rural areas or cities.

## CONCLUSIONS

This is the first report on *Bartonella* DNA among a large number of flea species in rodents that explored a gradient of urbanization across a wide geographic distribution in Chile. This paper adds seven new species to the list of fleas already reported to carry *Bartonella*. Although the prevalence of *Bartonella* DNA detected in this study was low, it is important to note that the villages and arid zone were the areas with the highest prevalence. In addition, the flea species that showed the highest *Bartonella* infection (*L. segnis* and *X. cheopis*) are fleas that have a wide distribution worldwide and are abundant in *R. rattus*. The other flea species collected in *R. rattus* corresponded to fleas that parasitize native rodents, which would indicate the degree of contact that these synanthropic rodents have with wild rodents, either directly or indirectly through the use of burrows, as they transmit parasites. This indicates that there are several species of *Bartonella* circulating in wild species. This finding is relevant, as parasite transmission could amplify bacterial infection among wild rodents, also increasing the probability with which infected fleas come into contact with humans in rural and wild areas. The results suggest the need to conduct further studies to verify whether these fleas might be transmitted to humans and cause disease.

## ACKNOWLEDGEMENTS

We thank to Nicole Inostroza, Elaine Monalize, and Maria Ignacia Najgle for their collaboration in collecting fleas. English-language editing of this manuscript was provided by Journal Prep Services.

### Funding

This work was supported by the National Fund for Scientific and Technological Development (FONDECYT) N° 11150875. The funders had no role in study design, data collection and analysis, decision to publish, or preparation of the manuscript.

## Grant Disclosure

The following grant information was disclosed by the authors:
National Fund for Scientific and Technological Development (FONDECYT): 11150875.

## Competing Interests

The authors declare that they have no competing interests.

## Author Contributions

- Lucila Moreno Salas conceived and designed the experiments, performed the experiments, analyzed the data, prepared figures and/or tables, approved the final draft.
- Mario Espinoza-Carniglia performed the experiments, analyzed the data, prepared figures and/or tables, approved the final draft.
- Nicol Lizama Schmeisser performed the experiments, approved the final draft.
- L. Gonzalo Torres contributed reagents/materials/analysis tools, approved the final draft, field work.
- María Carolina Silva-de la Fuente contributed reagents/materials/analysis tools, approved the final draft.
- Marcela Lareschi contributed reagents/materials/analysis tools, authored or reviewed drafts of the paper, approved the final draft.
- Daniel González-Acuña contributed reagents/materials/analysis tools, authored or reviewed drafts of the paper, approved the final draft.

## Animal Ethics

The following information was supplied relating to ethical approvals (i.e., approving body and any reference numbers):

The Comité de Ética de la Vicerrectoría de Investigación y Desarrollo de la Universidad de Concepción provided full approval for this research.

## Field Study Permissions

The following information was supplied relating to field study approvals (i.e., approving body and any reference numbers):

Field experiments were approved by Corporación Nacional Forestal (CONAF N°018-2015).

## DNA Deposition

The following information was supplied regarding the deposition of DNA sequences:

The sequences described here are available at GenBank: MK720786–MK720815.
The sequences are also available as a Supplemental File.

## Data Availability

Voucher specimens of each new association flea-host were deposited in the specimen repository of the Museo de Zoología, Universidad de Concepcion, Concepción, Chile (accession numbers: MZU-CCCC-46329–46336).

## Supplemental Information

Supplemental information for this article can be found online at http://dx.doi.org/10.7717/peerj.7371#supplemental-information.

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
