# Peer review of "Fleas of black rats (Rattus rattus) as reservoir host of Bartonella spp. in Chile"

_PeerJ, doi:10.7717/peerj.7371_

## Round 0.1 · original submission · Major Revisions

Dear Dr. Moreno Salas and colleagues:

Thanks for submitting your manuscript to PeerJ. I have now received two independent reviews of your work, and as you will see, the reviewers raised some concerns about the research. Despite this, these reviewers are optimistic about your work and the potential impact it will have on research communities studying Bartonella/flea/black rat vector biology. Thus, I encourage you to revise your manuscript accordingly, taking into account all of the concerns raised by both reviewers.

Please address minor English and grammatical problems.

Please also address why you elected to use novel primer sets versus those that are well-established in the community. Also, ensued that the text and figures/tables flow well together, with long-winded and unnecessary text culled.

While the concerns of the reviewers are relatively minor, this is a major revision to ensure that the original reviewers have a chance to evaluate your responses to their concerns.

I look forward to seeing your revision, and thanks again for submitting your work to PeerJ.

Good luck with your revision,

-joe

[]

·

Basic reporting

This is a descriptive study where authors detected DNA of Bartonella spp. in extracts of fleas collected on Rattus rattus. The structure and logical flow is there, however authors must pay more attention to the correct usage of the bacteria they are talking about (I assume authors talk about for example Bartonella koehlerae and Bartonella clarridgeiae). There are typos in many of the scientific names, which does not shed a good light on the rigor of the study. Overall there is evidence of understanding of the context of the flea/Bartonella knowledge. There are minor syntax and English problems. Sequence data are shared but I am puzzled by the length of DNA submitted. GenBank (BN: not GeneBank) accepts only sequences >200nt, but the suppl. sequences claimed to be submitted to GenBank are <200nt.

Experimental design

The sampling is commendably performed. I am not sure how authors assured that they trapped only Rattus rattus, based on my experience some native rodents will always be captured as well.When collecting fleas how did you assure that you captured all fleas on the rats?

Authors designed new sets of primers for both genes of Bartonella, why did they not use already established sets of primers? Anyhow, it is not clear how long are the fragments amplified esp if the comparison is based on <200bp. It would be desirable that longer fragments are produced.

How did authors prevent cross-contamination? Did you run blanks while isolating DNA?

Was sequencing done bidirectionally. It is highly desirable that it is done so.
It would be good that the alignment is included as suppl material.

Validity of the findings

The sequence length as outlined above needs to be clarified. If the generated sequences are >200nt it should be fine, however if not I would strongly suggest that authors consider amplifying more DNA. Consider protocols outlines in this: https://doi.org/10.1017/S0031182017001263

I found the discussion rather long for the amount of new data. Authors could be bit more succinct. Authors should refrain from repeating the results in discussion.

Additional comments

The references are in rather disorganized state, authors needs to pay more attention to the journal style and be consistent.

Reviewer 2 ·

Basic reporting

•The English language should be improved to ensure that a general audience can clearly understand your text. Major issues include run-on sentences and double-negatives (e.g., lines 53-59, lines 282-285). Minor issues include grammar, pluralization, spacing, and spelling errors throughout the manuscript (e.g., lines 45, 300, Tables 2-4).

•In the introduction (lines 53-53), there are more recent review articles that list Bartonella species detected in vectors for various mammalian hosts (e.g., Tsai et al. 2011., Cheslock & Embers. 2019.). The authors should cite the most recent work in the field. Also, Breitschwerdt & Kordick (2000) is not listed in alphabetical order in the references.

•The authors should consider reducing the amount of text in the discussion. Although I appreciate that they are providing information for each Bartonella species/strain, it is quite laborious to read through and takes away from the overall findings of the manuscript.

Experimental design

•There’s a disconnect between the introduction and the sample localities associated with hydrographic zones and seasonality. The introduction states the aim of the study is to investigate the “prevalence of Bartonella spp. in Rattus rattus fleas in areas with different human population density throughout Chile,” but makes no mention as to why the authors are also investigating different hydrographic zones and seasons. Please explain why these parameters were also examined.

•Please provide more details about the institutional animal care and use regulations (e.g., animal use was conducted in accordance with the Animal Welfare Act, protocol approval number, etc.).

•How were the rodents identified? Please reference taxonomic keys if used.

•Were the fleas collected immediately in the field or later in the lab? If later, were the rodents placed in separate bags or were they all placed together? My concern is that the prevalence and intensity data would be skewed if animals were placed together prior to ectoparasite removal. Please clarify.

Validity of the findings

•Some of the text in the results section does not correspond to the data presented in the tables. For example:
-Lines 188-189: A total of 179 fleas is listed in Table 2 when analyzed by locality (as opposed to the "177" mentioned in text). This number will change the data given in lines 188-189, 226-227.
-Lines 196-198: Plocopsylla wolffsohni is listed under the village locality too, not just in wild areas, according to Table 2.

•Fifteen species of fleas seems like a lot for R. rattus. Similar studies report one to five species of fleas (Loftis et al. 2006, Marie et al. 2006, Li et al. 2007, Reeves et al. 2007, Tsai et al. 2010). The authors do not address this finding in the manuscript. Also, are any of these species new host records? If yes, then there need to be voucher specimens.

•Please provide the confidence intervals for prevalence, intensity, and abundance values to indicate the accuracy of the estimation.

•In the abstract, the authors conclude that “villages in the arid zone correspond to areas with higher infection risk;” however, there are no statistical tests to support this claim in the results section (only prevalence of fleas was significantly higher in the villages, no stats presented in text on prevalence of Bartonella). Additionally, it states in the discussion that “no pattern is observed” between the prevalence of Bartonella in the different hydrographic zones.

•Again, the discussion states “in our study, a significantly higher prevalence of Bartonella DNA in the arid zone than in the other zone,” but there are no stats to back up this claim.

Additional comments

•The references need to be updated to reflect the articles cited in the text. For example, Zouari et al. (2017) is cited, but not listed in the reference section.

•Line 78: The common name for Pulex irritans is human flea, not dog flea.

•Line 82: Define “occupational risk”.

---

## Round 0.2 · accepted · Accept

Dear Dr. Moreno Salas and colleagues:

Thanks for revising your manuscript to PeerJ, and for addressing the concerns raised by the reviewers. I now believe that your manuscript is suitable for publication. Congratulations! I look forward to seeing this work in print, and I anticipate it being an important resource for research communities studying Bartonella/flea/black rat vector biology.

Thanks again for choosing PeerJ to publish such important work.

-joe

·

Basic reporting

ok

Experimental design

ok

Validity of the findings

ok

Additional comments

The manuscript was in general improved but remains rather long with rather hard to read and understand the details of the results.